# Effect of *N*-Acetylcysteine in Combination with Antibiotics on the Biofilms of Three Cystic Fibrosis Pathogens of Emerging Importance

**DOI:** 10.3390/antibiotics10101176

**Published:** 2021-09-27

**Authors:** Aditi Aiyer, Simone K. Visser, Peter Bye, Warwick J. Britton, Gregory S. Whiteley, Trevor Glasbey, Frederik H. Kriel, Jessica Farrell, Theerthankar Das, Jim Manos

**Affiliations:** 1Charles Perkins Centre, Infection, Immunity and Inflammation, School of Medical Sciences, The University of Sydney, Sydney, NSW 2006, Australia; aditi.aiyer@sydney.edu.au (A.A.); simonekvisser@gmail.com (S.K.V.); jessica.farrell@sydney.edu.au (J.F.); Jim.manos@sydney.edu.au (J.M.); 2Royal Prince Alfred Hospital, 12 Missenden Road, Sydney, NSW 2050, Australia; peter.bye@sydney.edu.au (P.B.); warwick.britton@sydney.edu.au (W.J.B.); 3Centenary Institute, The University of Sydney, Sydney, NSW 2006, Australia; 4Whiteley Corporation, Level 5, 12 Mount Street North Sydney, Sydney, NSW 2060, Australia; greg.whiteley@whiteley.com.au; 5School of Medicine, Western Sydney University, Sydney, NSW 2566, Australia; 6Whiteley Corporation, 19-23 Laverick Avenue, Tomago, NSW 2322, Australia; trevor.glasbey@whiteley.com.au (T.G.); erik.kriel@whiteley.com.au (F.H.K.)

**Keywords:** synergy, cystic fibrosis, biofilm, FICI, *Burkholderia*, *Stenotrophomonas*, *Achromobacter*, NAC

## Abstract

Cystic fibrosis (CF) is a genetic disorder causing dysfunctional ion transport resulting in accumulation of viscous mucus that fosters chronic bacterial biofilm-associated infection in the airways. *Achromobacter xylosoxidans* and *Stenotrophomonas maltophilia* are increasingly prevalent CF pathogens and while *Burkholderia cencocepacia* is slowly decreasing; all are complicated by multidrug resistance that is enhanced by biofilm formation. This study investigates potential synergy between the antibiotics ciprofloxacin (0.5–128 µg/mL), colistin (0.5–128 µg/mL) and tobramycin (0.5–128 µg/mL) when combined with the neutral pH form of *N*-Acetylcysteine (NAC_neutral_) (0.5–16.3 mg/mL) against 11 cystic fibrosis strains of *Burkholderia, Stenotrophomonas* and *Achromobacter* sp. in planktonic and biofilm cultures. We screened for potential synergism using checkerboard assays from which fraction inhibitory concentration indices (FICI) were calculated. Synergistic (FICI ≤ 0.5) and additive (0.5 > FICI ≥ 1) combinations were tested on irreversibly attached bacteria and 48 h mature biofilms via time-course and colony forming units (CFU/mL) assays. This study suggests that planktonic FICI analysis does not necessarily translate to reduction in bacterial loads in a biofilm model. Future directions include refining synergy testing and determining further mechanisms of action of NAC to understand how it may interact with antibiotics to better predict synergy.

## 1. Introduction

In patients with the genetic condition cystic fibrosis (CF), mutations in the cystic fibrosis transmembrane conductance regulator (CFTR) gene results in multi-organ dysfunction; however mortality is primarily attributed to progressive lung disease and respiratory failure [1]. The CFTR channel regulates chloride, bicarbonate, sodium, and water movement across epithelial surfaces. In the CF lung, ion transport is abnormal, resulting in accumulation of viscous and dehydrated mucus that fosters chronic biofilm-associated infection in the airways [1]. Biofilm matrices are composed of extracellular polymers (EPS) comprising microbially produced polysaccharides, extracellular-DNA and proteins [2]. Biofilm formation provides resistance to shear stress and tolerance to antibiotics in biotic environments [3], thereby promoting chronic infection. 

*Pseudomonas aeruginosa* remains a common colonising pathogen in patients with CF, however its prevalence and incidence are now decreasing, due in part to the widespread practice of aggressive antibiotic treatment to eradicate initial acquisition [4,5]. Other Gram-negative bacilli (GNB) of importance in the CF population include *Stenotrophomonas maltophilia, Achromobacter* spp. and the *Burkholderia cepacia* complex (BCC). *S. maltophilia* and *Achromobacter* spp. are becoming increasingly prevalent opportunistic pathogens in CF cohorts [1,6]. They have a reported overall prevalence of approximately 9% and 5%, respectively, in CF patients [7,8,9]. Although their clinical impact has been debated [8,10,11], more recently these organisms have both been associated with poorer outcomes in CF patients, including an increased risk of pulmonary exacerbations, death, and the need for lung transplantation [12,13,14]. The opportunistic GNB comprising the BCC have an overall prevalence of approximately 4% [4,7]. The BCC, specifically *B. cenocepacia*, have been associated with increased mortality, an accelerated decline in lung function and necrotising pneumonia [15,16]. Importantly, the presence of *B. cenocepacia* in airway cultures contraindicates lung transplantation [17].

These GNB display intrinsic resistance to multiple antibiotics classes, limiting treatment options even in the early stages of infection. They are also able to acquire resistance mechanisms and display biofilm growth [18,19,20], which further increases antibiotic resistance and tolerance. Although clinicians often use one or more of an aminoglycoside (tobramycin), a polymyxin (colistin), or a fluoroquinolone (ciprofloxacin) [21,22] to treat *P. aeruginosa,* the less common Gram-negative pathogens are variably susceptible to these antibiotics. Antibiotic effectiveness, however, may be enhanced if used in combination with a non-antibiotic compound.

*N*-Acetylcysteine (NAC), a thiol-based antioxidant and an endogenous precursor to glutathione, is currently used as an inhaled mucolytic [23]. Increasing evidence suggests it possesses both antibacterial and antibiofilm effects in vitro against many relevant respiratory pathogens [24,25,26]. NAC is also proteolytic and could inhibit bacterial adherence and EPS production [27] and play a key role in enhancing the effect of co-administered antibiotics. NAC has demonstrated antibiofilm activity with ciprofloxacin against *P. aeruginosa* [28] and has recently been shown to have antimicrobial and anti-biofilm effects against CF strains of BCC [26] and *S. maltophilia* [29] including synergy with colistin observed in the latter [29]. The effect of NAC on *Achromobacter* spp. biofilms has not been reported. Due to the increasing prevalence of this species and as polymicrobial infections are not uncommon in CF lung, the effect of the neutral pH form of NAC (NAC_neutral_) on *Achromobacter* spp. warrants investigation [9].

Synergy between compounds is defined as “when the minimal inhibitory concentration (MIC) of the individual compound is decreased significantly after the compounds are combined” [30,31]. An appropriate assessment of synergy would be advantageous in treatment decisions, especially as the multi-drug resistance profiles of these emerging species often leave clinicians with few therapeutic options. The acidic pH of intrinsic NAC causing airway irritation and bronchospasm impedes in vivo use [32], therefore NAC at neutral pH is used in treatment. This study investigates the potential for synergy between the antibiotics ciprofloxacin, colistin and tobramycin when combined with NAC_neutral_ against *B. cenocepacia,*
*S. maltophilia* and *A. xylosoxidans*. We aim to assess synergy of combinations against both planktonic and biofilm growth of these CF pathogens (Table 1).

## 2. Results

### 2.1. Neutralised NAC Is Ineffective against Planktonic Bacterial Growth

Planktonic growth reductions caused by NAC_neutral_ were assessed over 48 h. Percentages were calculated relative to the untreated controls at 100%. For all bacteria, regardless of strain, there were no concentrations of NAC_neutral_ alone that could inhibit visible growth of bacteria. The highest concentration tested was 16.3 mg/mL NAC_neutral_ which had varied reductions in planktonic growth across all strains. *A. xylosoxidans* strain 6908 was the only strain to have a significant decrease in growth at 16.3 mg/mL NAC_neutral_ with 81% growth compared to untreated control (*p* ≤ 0.05). For B cenocepacia ATCC 25608 and the clinical strains, M2167 and E5452, there was 30–48% growth while E5328 had the lowest growth of 5% compared to control; all reductions were significant (*p* ≤ 0.05). For *S. maltophilia* strains, a similar trend was observed as in *B. cenocepacia* strains, with significant 9–15% growth compared to the control across all strains (*p* ≤ 0.05) (Figure 1 and Table 2).

### 2.2. Effect of Antibiotics against Planktonic Bacterial Growth

Planktonic growth reductions by ciprofloxacin, colistin and tobramycin were assessed over 48 h. Percentage reductions were calculated as above (Section 2.1). In *A. xylosoxidans* ciprofloxacin-treated groups, all strains had a 5–9% growth at 16 μg/mL whereas at 32 μg/mL there was no growth which was significant (*p* ≤ 0.05). With *B. cenocepacia* ciprofloxacin-treated groups, all strains showed no growth at 64 μg/mL (*p* ≤ 0.05). In *S. maltophilia* strains, the ciprofloxacin MIC for ATCC 13637, 445595 and 19308 are 16-, 64-, 32 μg/mL, respectively (Table 2).

In *A. xylosoxidans* colistin treated groups, ATCC 2701 and 4365, no growth reductions were observed at the highest concentration, 128 μg/mL, with 14–18% growth. Strain 6908 and 6268 had MIC of 8 μg/mL and 32 μg/mL, respectively. For B. cenocepacia, all strains showed significant decreases in growth at 128 μg/mL with 73–78% growth compared to the control (*p* ≤ 0.05). The MIC for the other strains, ATCC 25608 and E5452 were 16- and 64 μg/mL, respectively, while other strains observed no significant decrease in growth at 128 μg/mL. In *S. maltophilia* colistin-treated groups, only strain ATCC 13637 observed total growth reduction while the two clinical strains at 32 μg/mL, 445595 and 19308, observed the same effect at 64 μg/mL (Table 2).

For all tobramycin treated groups, there was no significant decreases in growth compared to control except at the highest tested concentration (128 μg/mL) with 29–64% growth for *A. xylosoxidans*, 30–98% growth for *B. cenocepacia* and 0–62% growth for *S. maltophilia* strains. All reductions were significant compared to control (*p* ≤ 0.05), with the exception of M2167 (*p* > 0.05) (Figure 2 and Table 2).

### 2.3. Combination Therapy Demonstrated Synergism and Additive Effect against Planktonic Bacteria in Checkerboard Assays

Checkerboard assays were performed on all strains with combinations of NAC_neutral_ with ciprofloxacin, colistin and tobramycin in planktonic bacteria. Strain variations were observed across all bacteria. Synergy (FICI ≤ 0.5) of ciprofloxacin/NAC_neutral_ was observed for *B. cenocepacia* strains E5452 and E5328. Additive (0.5 > FICI ≥ 1) effects of ciprofloxacin/NAC_neutral_ were observed for *A. xylosoxidans* strains 6908 and 4365, *B. cenocepacia* strains E5452 and E532 and *S. maltophilia* strains 445595 and 19308.

Synergy of Colistin/NAC_neutral_ was seen for *A. xylosoxidans* strains ATCC 27601, 6908, and 4365 while additive effect was seen in all other strains. Colistin/NAC_neutral_ synergism was seen in *B. cenocepacia* ATCC 25608 only and additivism in both ATCC 25608 and E5452. *S. maltophilia* strains showed Colistin/NAC_neutral_ synergism or additivism in all strains. No synergistic or additive activity was observed of tobramycin/NAC_neutral_ against all bacteria tested (Figure 3).

### 2.4. Effect of Synergistic and Additive Combination Therapies against Irreversibly Attached Bacteria

The efficacy of synergistic and additive FICI combinations on irreversibly attached bacteria was tested to determine their minimum biofilm inhibitory concentration (MBIC) (Figure 4, Figure 5 and Figure 6). Bacterial growth percentages were calculated relative to 100% normalised untreated controls.

Reductions in irreversibly attached bacterial growth in *A. xylosoxidans* strains were assessed in Figure 4. Greater reductions in bacterial growth in all strains of *A. xylosoxidans* were observed using combinations of 32 μg/mL ciprofloxacin with any concentration (0.5–2 mg/mL) of NAC_neutral._ Growth ranged from 0.1–20%, a level significantly lower than that of the control (*p* ≤ 0.05) (Figure 4a–d). Conversely, higher concentrations of colistin in combination with NAC_neutral_ delivered lower reductions in bacterial growth. Combinations of 32 μg/mL colistin and 8.2 mg/mL NAC_neutral_ showed 44–67% bacterial growth compared to control across all strains (Figure 4e–f).

The effect of ciprofloxacin and NAC_neutral_ on irreversibly attached *B. cenocepacia* was strain dependent (Figure 5). In ATCC 25608 and E5328, all combinations of ciprofloxacin and NAC_neutral_ (8–32 μg/mL and 0.5–2 mg/mL, respectively) showed a significant 1.3–15.7% growth (*p* ≤ 0.05) (Figure 5a,d). In M2167, any concentration of ciprofloxacin with 1 mg/mL NAC_neutral_ recorded 40–48% growth (Figure 5b). Whereas strain E5452 had no combinations of ciprofloxacin/NAC_neutral_ that delivered growth reductions (Figure 5c). In colistin combinations, only high concentrations of NAC_neutral_ (8.2 mg/mL) with colistin (8–32 μg/mL) could deliver some reductions in bacterial growth, with approximately 32.5–63% growth for all strains except for E5452. (Figure 5e–h).

The effect of both ciprofloxacin and colistin in combination with NAC_neutral_ on irreversibly attached *S. maltophilia* was assessed in Figure 6. In ATCC 25608 and E5328, all combinations of ciprofloxacin and NAC_neutral_ (8–32 μg/mL and 0.5–2 mg/mL, respectively) showed a 0.1–16.3% significant growth compared to control (*p* ≤ 0.05) (Figure 6a–c). The specific combinations of NAC_neutral_ and colistin, 8.2 mg/mL and 8 μg/mL, respectively, were the only combinations that could reduce growth below 50%; showing a growth of 16–36% compared to control (Figure 6d–f).

### 2.5. Time Course Bacteriostatic Effect of Synergistic and Additive Combination Therapies

The potential bacteriostatic or bacteriocidal effect of the combination therapies over 96 h was assessed by growing bacteria in the presence of treatments, removal of treatments at 48 h and assessing regrowth over time. Combinations were compared visually to the orange untreated control line. The removal of treatments resulted in regrowth of strains, indicated by an increase in OD_600nm_ comparable to the level of the untreated control. While no combinations resulted in reduction in OD_600nm_ to 0 or were bactericidal, there were some combinations (Table 3) that had lower, significant regrowth of bacteria than others at the final timepoint of 96 h. All comparisons are referring to this 96 h timepoint. 

In the case of *A. xylosoxidans* ATCC 27601 (Appendix A), significant differences (*p* ≤ 0.05) were observed between the OD_600nm_ of the untreated control and combinations (J) and (L). In strain 6268, the untreated control was significantly different to (L). Similarly in strain 6908 the combinations (B) and (G), were significantly different from the untreated control (*p* ≤ 0.05).

Two of the three clinical *B. cenocepacia* strains had significant differences (*p* ≤ 0.05) between the OD_600nm_ of the untreated controls and specific combinations (Appendix A). The untreated control for M2167 was significantly higher than (D), (J) and (L). In E5452, the untreated control was significantly higher compared to (L) (*p* ≤ 0.05).

All *S. maltophilia* strains observed significant decreases in OD_600nm_ between the untreated controls and certain combinations (Appendix A). Combinations (C), (D), (F) and (L) were significantly lower than the OD_600nm_ of the untreated ATCC 13637 control (*p* ≤ 0.05). Both clinical strains showed significant differences between (L) and their untreated controls. From strain 445595, (L) had a lower OD_600nm_ than the untreated control. In strain 19308, the untreated control was significantly higher than the (L) combination (*p* ≤ 0.05). 

### 2.6. Effect of Synergistic and Additive Combination Therapies against Mature Biofilms In Vitro

Figure 7, Figure 8 and Figure 9 show the antibiofilm activity of NAC_neutral_ with ciprofloxacin and colistin combinations on all 11 strains of bacteria. Mature biofilms, grown for 48 h, were exposed to 12 different combinations (Table 3) for 24 h. Viable cell counts after treatment were determined using colony forming units (CFU/mL). Statisical analyses were performed by comparing combinations against both the antibiotic and NAC_neutral_, and significance was only reported if combination treatment had a significantly different effect (*p* ≤ 0.05) to both components.

The CFU/mL from mature biofilm experiments were also compared to results from planktonic synergy analysis (FICI) (presented in Figure 3). (✓) indicates that the combination tested corresponds to planktonic result in FICI analysis while (✖) indicates that the combination tested does not correspond to planktonic result in FICI analysis.

The untreated viable cell counts for each bacteria in our biofilm model ranged from 2 × 10^9^–2.4 × 10^9^ CFU/mL for *A. xylosoxidans* and *B. cenoepacia* and ranged from 1.8 × 10^8^–2.4 × 10^9^ CFU/mL for *S. maltophilia*. Combination treatment did not result in consistently lower viable cell counts than treatment with either component alone, therefore synergy or additive effects were not seen when treating mature biofilms; moreover a strain dependent effect was observed (Figure 7, Figure 8 and Figure 9).

Figure 7 shows the effect of all combinations on *A.xylsoxidans* strains. In ATCC 27601, combination (D) recorded indifference in FICI analysis however reported a significant difference in growth of 1–2 log_10_ CFU/mL to both 64 μg/mL ciprofloxacin and 1 mg/mL NAC_neutral_ (*p* ≤ 0.05). Combinations (A) and (B) recorded indifference in FICI analysis and these results corresponded to the CFU/mL (Figure 7a). Clinical strain 6268 reported combination (K) as synergistic in FICI analysis and this corresponded with a significant decrease in CFU/mL of 2.5 log_10_ CFU/mL (*p* ≤ 0.05). Combination (J) reported a large 2–3 log_10_ decrease in CFU/mL in comparison to both 16 μg/mL colistin and 8.2 mg/mL NAC_neutral_, this result contrasts to the FICI analysis which suggest only an additive effect and thus a smaller reduction. However, combinations (A), (D), (F), (G), (I), and (J) did not correspond to their reported FICI. Combinations (F), (G) and (J) were not killed in planktonic phase, but reported 1–2 log_10_ CFU/mL reductions in biofilms. While (A), (D) and (I) showed no reduction in mature biofilms that were synergistic in planktonic phase (Figure 7b). In strain 4356 only two combinations, (H) and (I), matched their synergy classification recording significant 1–4 log_10_ CFU/mL reductions (*p* ≤ 0.05). Only one combination (B) reported significant killing despite not matching its FICI classification where planktonic killing was not observed (Figure 7d).

Figure 8 shows the effect of all combinations on *B. cenocepacia* strains where no combinations matched FICI synergism but were able to successfully reduce CFU/mL for any strains. In ATCC 25608, combination (L) showed no significant difference between the controls, corresponding to indifference in FICI. Combination (D), however, showed significant reduction of 1 log_10_ CFU/mL but this did not correspond to planktonic killing in FICI testing (Figure 8a). Clinical strains M2167 and E5328 similarly experienced significant decreases in CFU/mL, approximately 1–2 log_10_ (*p* ≤ 0.05) depending on the combination and this did not correspond to the results from their FICI results, where no planktonic killing was observed at those concentrations (Figure 8b,d).

Figure 9 shows the effect of all combinations on *S. maltophilia* strains where significant decreases in CFU/mL were observed, but again, not all matched to their FICI analysis performed on planktonic cultures. In ATCC 13637, combination (J) was additive in planktonic cultures and a corresponding effect was observed in mature biofilms, with a decrease of 1–2 log_10_ CFU/mL in comparison to controls (Figure 9a). While combinations (D), (E) and (F) did not match to their FICI classifications of antagonism and no planktonic killing they were still able to deliver a reduction in log_10_ CFU/mL of 1−2 depending on the combination (Figure 9a). Combinations (B), (C) and (L) when tested in 445595, demonstrated no killing in planktonic FICI testing, however reductions of 2 −3 log_10_ CFU/mL were observed depending on the combination (Figure 9b). Strains 445595 and 19308 both had CFU/mL reductions with combination (D) of 1 log_10_ CFU/mL (Figure 9b,c).

## 3. Discussion

The Gram-negative bacteria *A. xylosoxidans*, *B. cenocepacia*, and *S. maltophilia*, are of emerging importance in CF. The prevalence of *A. xylosoxidans* and *S. maltophilia* have been increasing and while *B. cencocepacia* has been slowly decreasing, its presence contraindicates lung transplantation and therefore effective treatment options are needed [17]. Due to multi drug resistance and biofilm growth characteristics, a combination therapy (CT) with NAC and antibiotics has been investigated.

Our goal was to use the concept of synergy, an assessment of antimicrobial combinations to determine whether the effect of the two antimicrobials was greater than the sum of their individual activities [33]. We could then utilise any combinations which had either a synergistic or additive effect to treat mature biofilms. The combinations utilised are as shown in Table 3. Although we found evidence of synergy between NAC and antibiotic in FICI analysis in planktonic culture, these results obtained in this planktonic-based test were mostly not confirmed in testing of viable cell counts (CFU/mL) from mature biofilm. Intriguingly, there were more outcomes that did not correlate with FICI analysis (Figure 3, Figure 7, Figure 8 and Figure 9). CFU/mL reductions were observed in some strains for combinations that either showed antagonism in planktonic cultures or were not able to kill planktonic bacteria. Furthermore, for the few combinations that had both a synergistic classification according to FICI and were able to reduce viable cell count in biofilms, the effect was not consistent across all strains. As in other studies of antibiotic resistance and synergy, our results were strain and concentration dependent [26,29].

Statistical analyses were performed on CFU/mL cultures with this in mind. Our statistical question was not only, “did the CT perform better than the antibiotic, alone” but also “did the CT perform better than each of its two components”. Currently NAC is used as a nebulised mucolytic agent. Some studies investigating NAC for the treatment of *Burkholderia* spp. and *S. maltophilia* in vitro [26] have found a reduction in bacterial load when the antioxidant is used alone. We were able to see similar effects, through strain-to-strain variability in tested bacteria. Thus, it is important that if any CT is employed using these two components it should perform better than NAC or the antibiotic alone, especially in the current landscape of emerging resistance to antimicrobials.

Ciprofloxacin, colistin and tobramycin were chosen for this study. The reason these antibiotics were chosen relates mainly due to their ability to be delivered via nebulisation [1,34,35], although they are more commonly used against *P. aeruginosa* infection in CF. Two other antibiotics that are regularly used in therapy against these less common Gram-negative organisms are trimethoprim-sulfamethoxazole and doxycycline, however they were not included as they are currently only available as an oral formulation. Given the prevalence of all the above species in mixed Gram-negative infection in CF [7], it would also be useful to assess whether the effect of these antibiotics could be enhanced when used together with NAC.

Ciprofloxacin, a fluoroquinolone, inhibits DNA replication by inhibiting bacterial DNA topoisomerase and DNA-gyrase and is most effective against *P. aeruginosa* infections [22]. It has been studied in combination with NAC in its intrinsic acidic pH form and has shown some reductions in CFU/mL. In a previous study by our group, we showed that acidic NAC in combination with the MIC of ciprofloxacin was able to reduce the bacterial loads more than the antibiotic alone in *B. cenocepacia* isolates [36]. In this former study we concentrated on combinations with acidic NAC and their synergism was not fully investigated. Therefore, we recognize that this combination and its effects could be attributed to the overall acidity of NAC. We chose to focus on ciprofloxacin in our current study, but in combination with NAC_neutral_, which is similar to the pH of proprietary versions of NAC that are approved for use in the lung [37,38].

Our second antibiotic choice, colistin, a polymyxin antibiotic, disrupts the outer cell membrane causing leakage of intracellular contents and bacterial death [39]. It is used frequently in CF patients as a nebulised antibiotic to treat *P. aeruginosa* [40]. Additionally, BCC bacteria have also displayed a high level of intrinsic resistance to polymyxin antibiotics including colistin [39]. Despite this, in a study by Ciacci et al. (2019), was able to show antibiofilm synergistic activity of colistin/NAC against *S. maltophilia* [29]. They also reported there was a natural variation in reported synergy results between the strains they tested, which were from intra-abdominal, lower respiratory tract, bloodstream, and CF infections. Our study focused purely on CF isolates, and we included three bacterial species in our study to ascertain if similar effects of synergy could be observed across these bacteria.

Our third choice, tobramycin, an aminoglycoside, binds to bacterial ribosomal subunits and inhibits bacterial protein synthesis [41]. Nebulised tobramycin is also commonly used in the treatment of *P. aeruginosa* in CF patients, however in recent years, there has been a pattern of emerging resistance of CF bacteria against aminoglycoside antibiotics [42]. Nevertheless, in a study by Kennedy et al. (2015), it was shown that the thickness of *B. cenocepacia* biofilms were able to be reduced despite attenuated killing. Thus, while they are still used in treatment, aminoglycoside efficacy is usually enhanced with other antibiotics, or by administration of high doses [43]. In this study it was selected to determine whether the effect of tobramycin is enhanced when used in combination with NAC_neutral_.

Synergy reporting is not currently performed in routine susceptibility testing, and the potential benefit of its use is that we are able to consider any positive interactions between certain combinations of compounds. Doern (2014) highlighted that synergy reporting does depend greatly on the methodology used, which implies there is no gold standard for testing. Furthermore, this review also highlighted that it is difficult to link the effects of synergistic compounds to patient outcomes [30,33]. A potential reason for this is the discrepancy between the phases of bacterial growth between planktonic and biofilm [44]. Planktonic bacteria are designed to colonize new niches whereas biofilms are designed to protect bacteria as they reproduce. This shift from planktonic to biofilm modes of growth are known to be controlled by complex regulatory networks and result in biofilms exhibiting low metabolic activity alongside strict quorum sensing guidelines [44]. This may explain the increased tolerance against antibiotics seen in biofilms and the difficulties in using a planktonic screening method to assess susceptibilities of colonising bacteria within a biofilm.

A limitation of synergy testing is that it does not account for the individual mechanisms of action (MOA) of the two compounds being tested. By understanding the MOA of tested compounds, it may influence which candidates are chosen for testing. Much work has been performed on ascertaining the MOA of antibiotics however the same cannot be said for NAC. NAC, a cysteine prodrug, is currently used as a mucolytic [45] and can be tolerated in high doses [37]. However, its MOA is yet to be fully elucidated both individually and in the context of combination therapy. Currently it is thought to work in three main ways: breaking disulphide bonds in mucus or proteins in biofilm matrices [46], as a reactive oxygen species scavenger and as a precursor to glutathione synthesis [47]. These mechanisms may not be generalised to explain NAC’s effects in other circumstances, such as in bacterial biofilms. Blasi et al. (2016) hypothesised that NAC interferes with the intracellular redox equilibrium via its free thiol group [25]. This could have downstream effects on bacterial cell metabolism, and possibly account for inhibition of biofilm formation [48] and disruption of its matrix cytoarchitecture via chelation of divalent cations such as calcium or magnesium.

Thus, a future direction would involve determining the MOA of NAC more clearly, as this may provide further insight into potential for synergistic activity and guide further antibiotic choices to test in combination. For synergy testing to be more clinically relevant, it must take into account the different phases of bacterial growth in the CF lungs by assessing synergistic effects against biofilms in addition to planktonic cultures. Consideration could also be given to adapting and standardising the testing methodology for biofilms in addition to planktonic growth. Standardising synergy testing methodology is warranted so that results are reproducible across different laboratories. This would enhance clinical relevance when determining appropriate therapeutic intervention.

## 4. Materials and Methods

### 4.1. Cystic Fibrosis Opportunist Strains Tested

*B. cenocepacia* ATCC 25608™, *S. maltophilia* ATCC 13637™, *A. xylosoxidans* ATCC 27601™ (American Type Culture Collection, Manassas, VA, USA), three clinical *B. cenocepacia* (Microbiology Department, Westmead Hospital, Sydney, Australia), three *A. xylosoxidans* and two *S. maltophilia* CF strains isolated from patient sputum cultures (CF clinic, Royal Prince Alfred Hospital, Sydney, Australia) were stored at −80 °C. One author, Simone Visser, was involved in sputum collection from patients and the isolation of *A. xylosoxidans* and *S. maltophilia* from patient’s sputa. No authors were involved in collection or isolation of *B. cenocepacia* strains. All clinical strains were de-identified. Isolate susceptibility was determined at the Microbiology Department, Royal Prince Alfred Hospital (RPAH), Sydney using different CLSI approved methods detailed in Table 1.

### 4.2. Culture Conditions and Preparations of Treatments

All isolates were grown in Tryptone soya broth (TSB) and on Tryptone soya agar (TSA) plates (ThermoFisher Scientific, Sydney, Australia) for 48 h (37 °C and 150 rpm). “Diluted bacterial cultures” referred to herein are maintained at OD_600nm_ 0.1 ± 0.02. NAC, ciprofloxacin, colistin and tobramycin stock solutions were prepared immediately before use by dissolving the respective powder (Sigma-Aldrich, Sydney, Australia) in autoclaved, deionised water and in the case of ciprofloxacin, 0.1 M HCl. All experiments were performed in TSB or phosphate-buffered saline (PBS: 137 mM NaCl, 2.7 mM KCl and 10 mM phosphate, pH 7.41) (POCD Healthcare, Sydney, Australia). All experiments were performed in biological triplicate.

### 4.3. Preparation of NAC Medium

NAC stock solutions (97.9 mg/mL) (600 mM) were prepared immediately before use. NAC powder was dissolved in sterile, autoclaved TSB, pH was adjusted to 6.5–7.4 with 0.1 M NaOH, and the solution was filtered through a 0.22 μm membrane filter. pH-adjusted NAC will be referred to herein as “NAC_neutral_”.

### 4.4. Preparation of Antibiotics

Ciprofloxacin, colistin and tobramycin stock solutions (900 μg/mL) were prepared immediately before use. Ciprofloxacin was specifically required to be dissolved in 0.1 M HCl (pH = 4.0 ± 0.2). All solutions were filtered through a 0.22 μm membrane filter.

### 4.5. Effect of Antioxidant and Antibiotics against PLANKTONIC Bacteria

MIC was defined as the lowest concentration of antimicrobial to inhibit planktonic growth, as determined by the broth microdilution method [36,49]. The effects of the antioxidant and antibiotics on isolates were determined by inoculating diluted bacterial cultures in TSB into 96-well plates (Corning Corp., New York, NY, USA). Where indicated, TSB media was spiked with antibiotics to a final concentration of 0-, 2-, 4-, 8-, 16-, 32-, 64- and 128 μg/mL or similarly, with NAC_neutral_ at 0-, 1-, 2-, 4.1-, 8.2-, 16.3 mg/mL. Plates were then incubated for 48 h at 37 °C and 100 rpm. Absorbance (OD_600nm_) was recorded using a plate reader (Tecan infinite M1000 Pro, Switzerland). Treatments measured percentage decrease in bacterial growth with respect to untreated controls (100% growth).

### 4.6. Synergy Testing via Checkerboard Assays

Synergy susceptibility testing was performed using the broth micro-diliution method as described previously [50]. Briefly, checkerboard synergy testing was performed in triplicate using 96-well microplates (Corning Corp., New York, NY, USA). Positive growth controls were performed on the same plate in wells not containing antimicrobials. Positive, negative and single treatment controls were also assayed on the same microplate. Combinations tested used 0-, 0.5-, 1-, 2-, 4-, 8-, 16-, 32-, 64-, and 128 μg/mL of all antibiotics against 0-, 0.5-, 1-, 2-, 4.1-, 8.2-, and 16.3 mg/mL of NAC_neutral_. Tests were performed in triplicate for each bacterial strain tested. Microtitre plates were incubated for 24 h at 37 °C in 150 rpm shaking conditions and turbidity was assessed visually using the Tecan600 Microplate reader at OD_600nm_ absorbance. Interpretation of checkerboard assay follows previously described method [51]. In brief, the Mean Fractional Inhibitory Concentration (FIC) index was calculated using the concentrations in the first non-turbid (clear) well found in each set of wells with negative individual growth controls. The concentrations across replicates were then averaged [50]. The bounds of synergy have been set as follows: ≤0.5, synergy; > 0.5 ≥ 1, additive; >1–4, indifference; >4, antagonism. FICI was calculated for all treatment combinations tested.

### 4.7. Effect of Antioxidant and Antibiotics on Prevention of Biofilm Maturation Following Irreversible Bacterial Attachment

The MBIC of individual antibiotics and antioxidants on irreversibly attached bacteria were determined using previously detailed methods with modifications [36,52]. Briefly, the MBIC values for bacteria were determined using 200 μL of diluted bacterial culture added to 96-well plates (Corning Corp., New York, NY, USA) and incubated for 2 h at 37 °C (100 rpm), for irreversible bacterial attachment. The supernatant was removed, and the plates washed twice with PBS. Plates were incubated for a further 48 h in the presence of the antibiotic or antioxidant dissolved in TSB. Absorbance (OD_600nm_) was recorded using a plate reader (Tecan infinite M1000 Pro). Treatment groups were compared for percentage decrease compared to TSB growth control showing 100% growth.

### 4.8. Time-Kill Assay of Single versus Combination Treatments against Planktonic Cultures

Time-kill bactericidal activity assays were performed on all tested strains, as described previously [36]. The bacteriostatic or bactericidal effect of NAC, defined as termination or killing of bacteria, respectively, was determined by inoculating TSB-diluted cultures in 96-well flat-bottomed plates in the presence of final concentrations in Table 3. Plates were then incubated at 37 °C in a shaking incubator (100 rpm) for 48 h, with absorbance (OD_600nm_) measured at various timepoints. Following incubation, plates were centrifuged at 4500× *g* for 15 min, the supernatant discarded, and plates washed twice with PBS. Plates were then centrifuged again at 4500× *g* for 15 min, and wells were refilled with fresh TSB. Growth of bacteria was monitored by measuring absorbance for a further 48 h; for a total of 96 h. Percentage bacterial revival was calculated to determine bacteriostatic or bactericidal effect.

### 4.9. In Vitro Mature Biofilm Susceptibility Testing with Single and Combination Treatments Using Colony Forming Units (CFU/mL)

Mature biofilms were grown for 48 h and subsequently treated for 24 h with combinations highlighted in Table 3 for 24 h as described above. After washing once with PBS, 200 μL PBS was added and wells were scraped with a pipette tip 10 times horizontally and vertically. The scraped biofilm was then thoroughly homogenised by pipetting up and down five times. To establish a CFU count, a Whitley Automatic Spiral Plater (WASP) was used (Don Whitley Scientific, West Yorkshire, UK) where the WASP automatically plated 50 μL of diluted bacterial suspension. Plates where then incubated for 48 h at 37 °C. Following incubation, the plate colonies were enumerated and expressed as CFU/mL.

### 4.10. Statistical Analysis

Statistical analysis was performed using GraphPad Prism version 6.0 (San Diego, CA, USA). Student *t*-tests were performed on all the data. MBIC data (Figure 4, Figure 5 and Figure 6) required one sample *t*-tests and Wilcoxon rank testing because of the layout of data, where samples were compared to the normalised control mean of 100%. D’Agostino-Pearson and Shapiro–Wilk normality tests were applied. Multiple comparison tests were performed by Kruskal–Wallis test with Dunn’s correction. Results are considered statistically significant if *p* < 0.05 (*****).

## 5. Conclusions

In conclusion, our study suggests that planktonic FICI analysis does not necessarily translate to a reduction in bacterial loads in a biofilm model. We observed significant decreases in planktonic and attached bacterial growth using combinations of ciprofloxacin/colistin and NAC_neutral_ that were synergistic. We also observed instances of significant reduction in CFU/mL in mature biofilms that were not correlated to planktonic synergy results. We believe our results in this study to be a springboard for further assessment of the efficacy of NAC in the treatment of CF, not just as a mucolytic but potentially in reducing bacterial loads.

We propose that future directions include determining the mechanism of action of NAC to understand how it may interact with antibiotics, and thus better predict synergy. While improving patient outcomes is the long-term goal, it is important initially to develop a clearer methodology that can be verified in different bacterial growth phases and eventually in replicable cell and animal models.

## Figures and Tables

**Figure 1 antibiotics-10-01176-f001:**
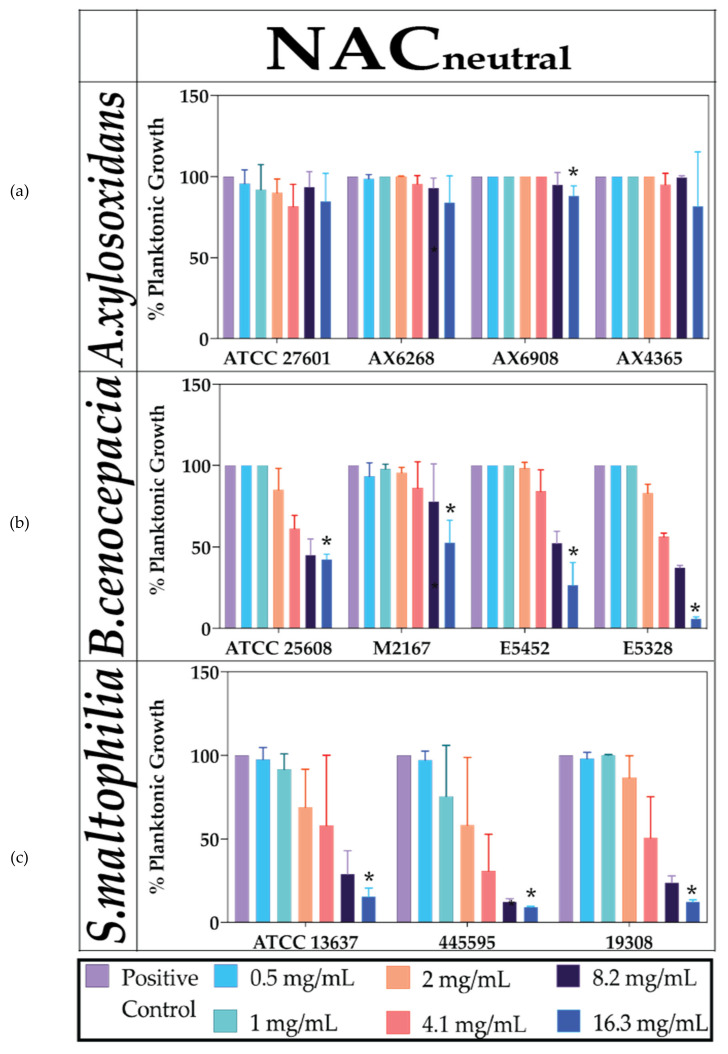
The effect of neutralised NAC alone on planktonic bacterial growth. Concentration-dependent effect of NAC_neutral_ on percentage planktonic growth of (**a**) *B. cenocepacia* (**b**) *A. xylosoxidans* and (**c**) *S. maltophilia* strains. Statistical analyses were performed using unpaired Student’s *t*-tests (with Welch’s correction) and multiple comparison tests (Kruskal–Wallis test with Dunn’s correction). Significance cut-offs are as follows *p* ≤ 0.05 (*****). Data represent an average of *n* = 3 biological replicates.

**Figure 2 antibiotics-10-01176-f002:**
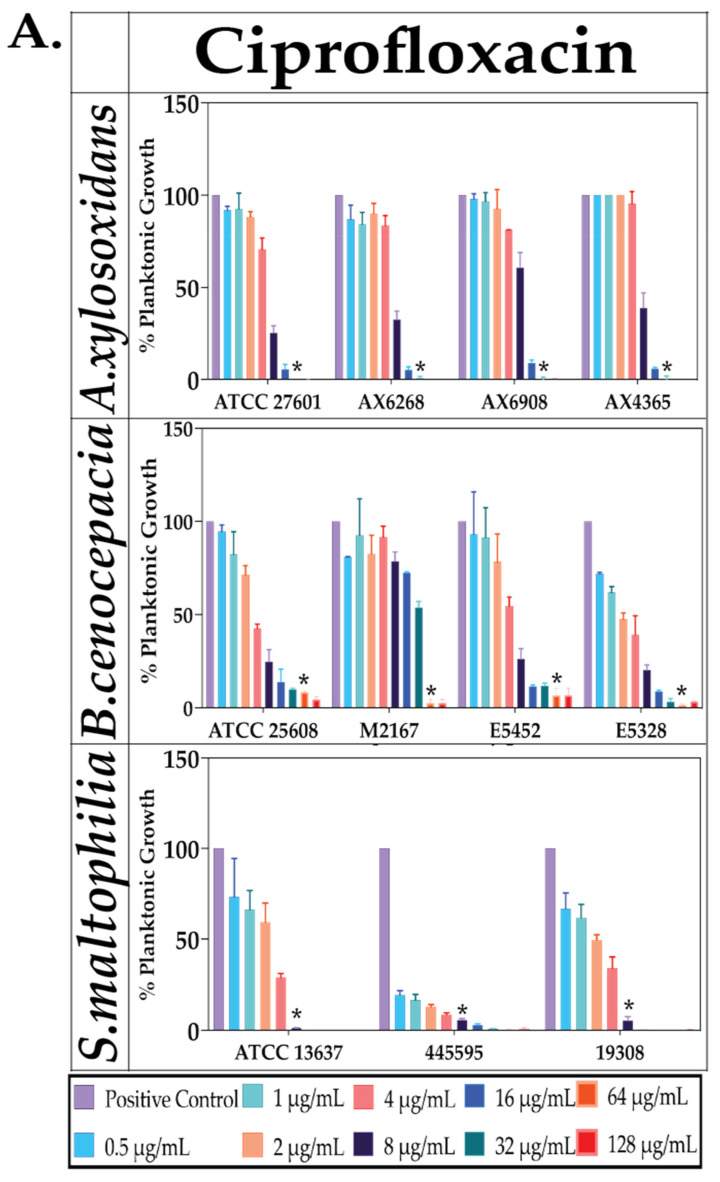
The effect of antibiotics alone on planktonic bacterial growth. Concentration-dependent effect of the antibiotics (**A**) ciprofloxacin, (**B**) colistin and (**C**) tobramycin on percentage planktonic growth of *B. cenocepacia, A. xylosoxidans* and, *S. maltophilia* strains. Statistical analyses were performed using unpaired Student’s *t*-tests (with Welch’s correction) and multiple comparison tests (Kruskal–Wallis test with Dunn’s correction). Significance cut-offs are as follows *p* > 0.05 (*n*), *p* ≤ 0.05 (*****). Data represent an average of *n* = 3 biological replicates.

**Figure 3 antibiotics-10-01176-f003:**
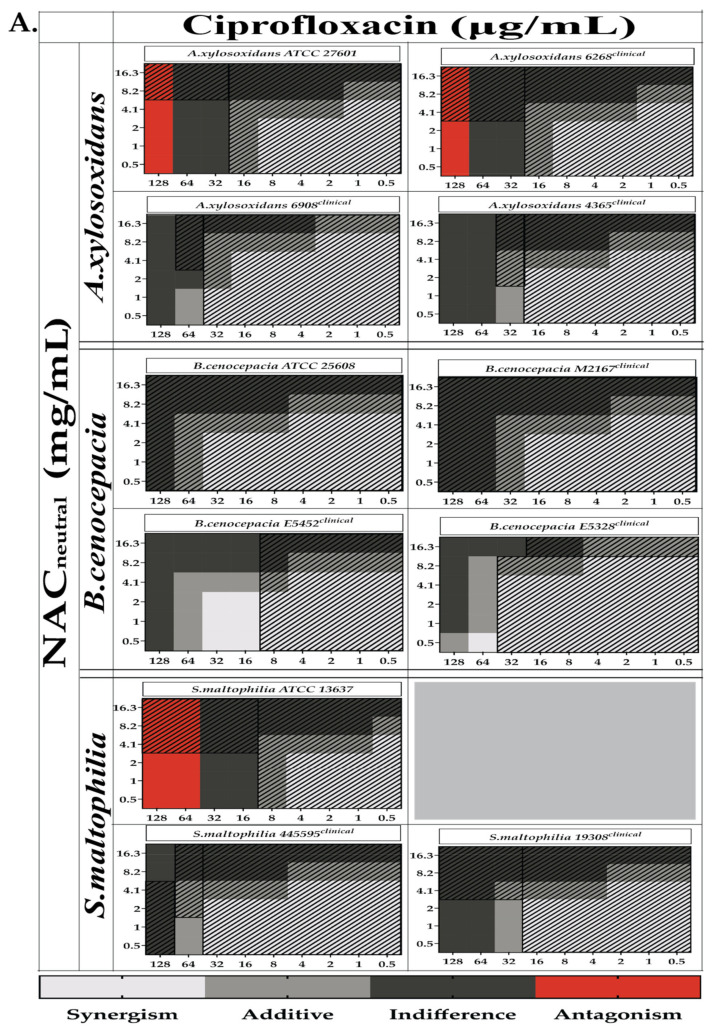
In vitro synergism of NAC_neutral_/antibiotic combinations against bacterial strains grown in planktonic phase. Heat map representing color-coded fractional inhibitory concentration index (FICI) as calculated from planktonic growth of all bacterial strains of *A. xylosoxidans*, *B. cenocepacia*, *S. maltophilia* against combinations of (**A**) ciprofloxacin, (**B**) colistin and (**C**) tobramycin in combination with NAC_neutral_. Shaded regions represent areas where combinations were ineffective in killing planktonic bacteria. The bounds of synergy have been set as follows: ≤0.5, synergy; 0.5 > FICI ≥ 1, additive; >1–4, indifference; >4, antagonism.

**Figure 4 antibiotics-10-01176-f004:**
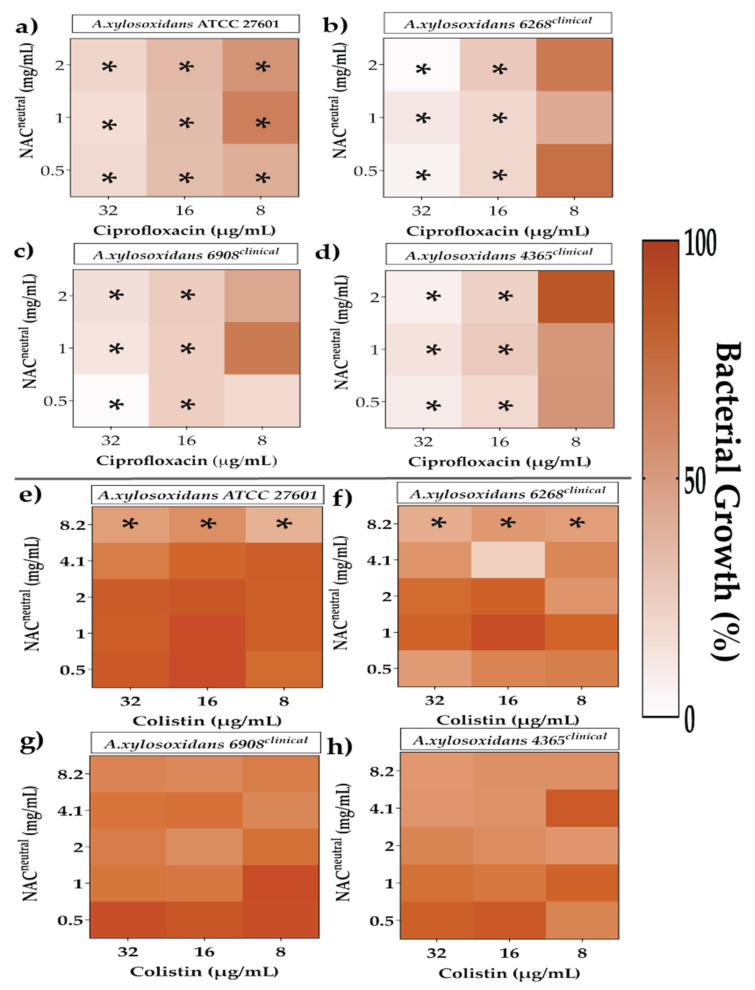
The effect of chosen synergistic and additive combinations against irreversibly attached *Achromobacter xylosoxidans*. Heat map representing greyscale-coded planktonic bacterial growth where black represents 100% bacterial growth and white represents 0% irreversibly attached planktonic bacterial growth. Irreversibly attached bacterial growth has been measured in the presence of ciprofloxacin (**a**–**d**) and colistin (**e**–**h**) in combination with NAC_neutral_ for *A. xylosoxidans* strains. Statistical analyses were performed using one sample *t*-test and Wilcoxon rank tests. Significance cut-offs are as follows *p* ≤ 0.05 (*****). Data represent an average of *n* = 3 biological replicates.

**Figure 5 antibiotics-10-01176-f005:**
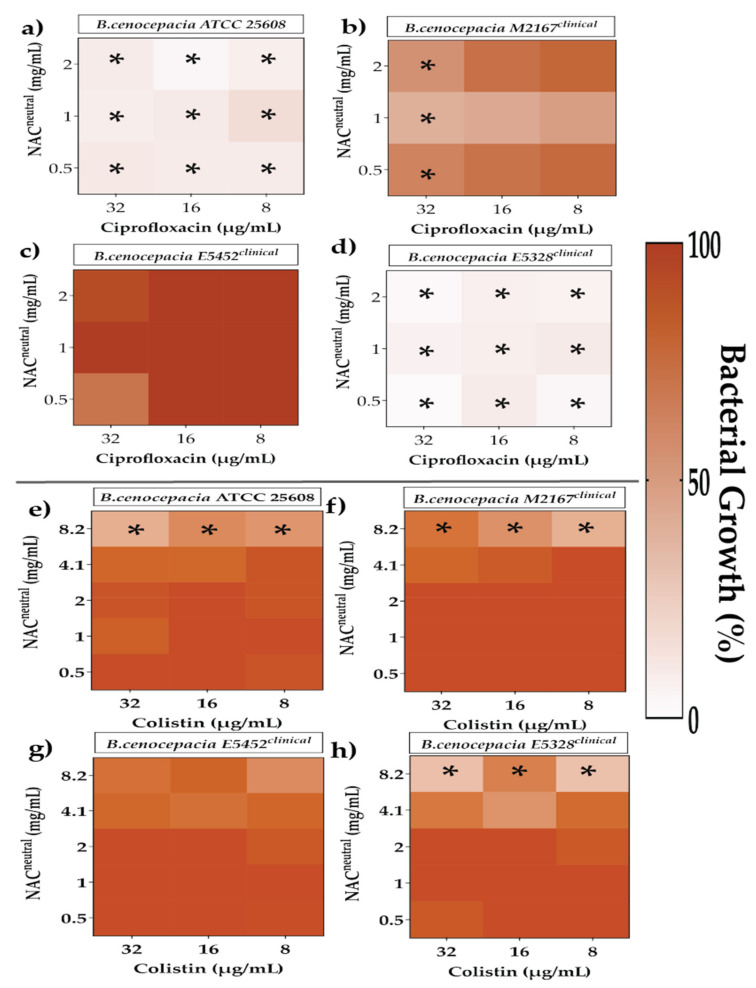
The effect of chosen synergistic and additive combinations against irreversibly attached Burkholderia cenocepacia. Heat map representing greyscale-coded planktonic bacterial growth where black represents 100% bacterial growth and white represents 0% irreversibly attached planktonic bacterial growth. Irreversibly attached bacterial growth has been measured in the presence of ciprofloxacin (**a**–**d**) and colistin (**e**–**h**) in combination with NAC_neutral_ for *B. cenocepacia* strains. Statistical analyses were performed using one sample *t*-test and Wilcoxon rank tests. Significance cut-offs are as follows *p* ≤ 0.05 (*****). Data represent an average of *n* = 3 biological replicates.

**Figure 6 antibiotics-10-01176-f006:**
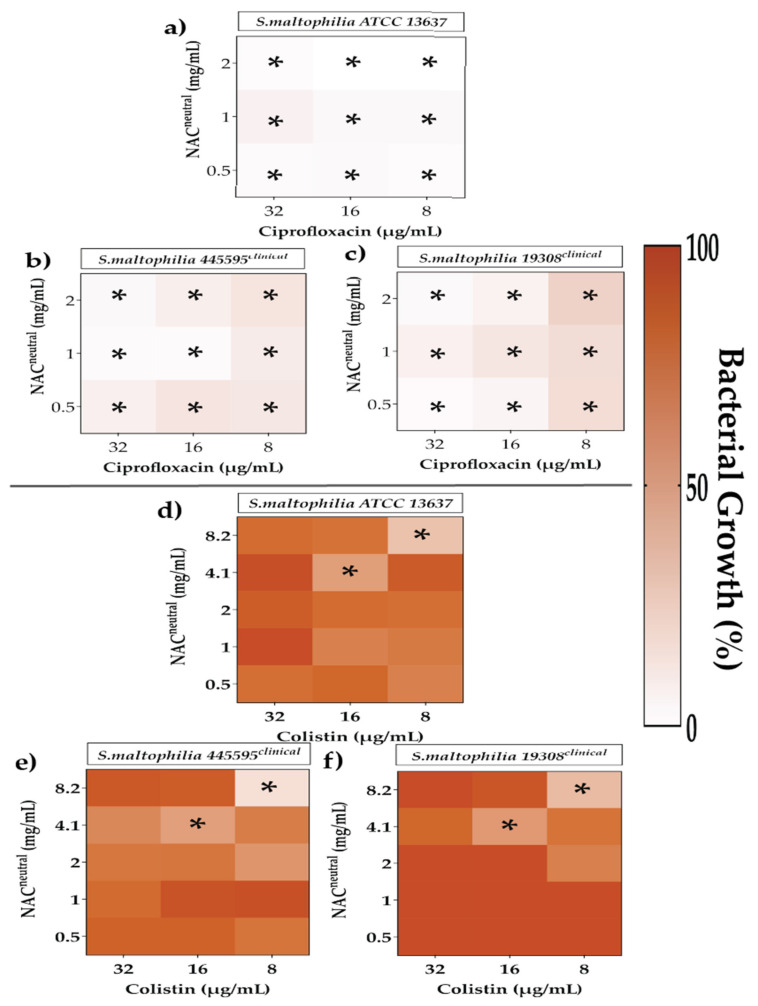
The effect of chosen synergistic and additive combinations against irreversibly attached *Stenotrophomonas maltophilia*. Heat map representing greyscale-coded planktonic bacterial growth where black represents 100% bacterial growth and white represents 0% irreversibly attached planktonic bacterial growth. Irreversibly attached bacterial growth has been measured in the presence of ciprofloxacin (**a**–**d**) and colistin (**e**,**f**) in combination with NAC_neutral_ for *S. maltophilia* strains. Statistical analyses were performed using one sample *t*-test and Wilcoxon rank tests. Significance cut-offs are as follows *p* ≤ 0.05 (*****). Data represent an average of *n* = 3 biological replicates.

**Figure 7 antibiotics-10-01176-f007:**
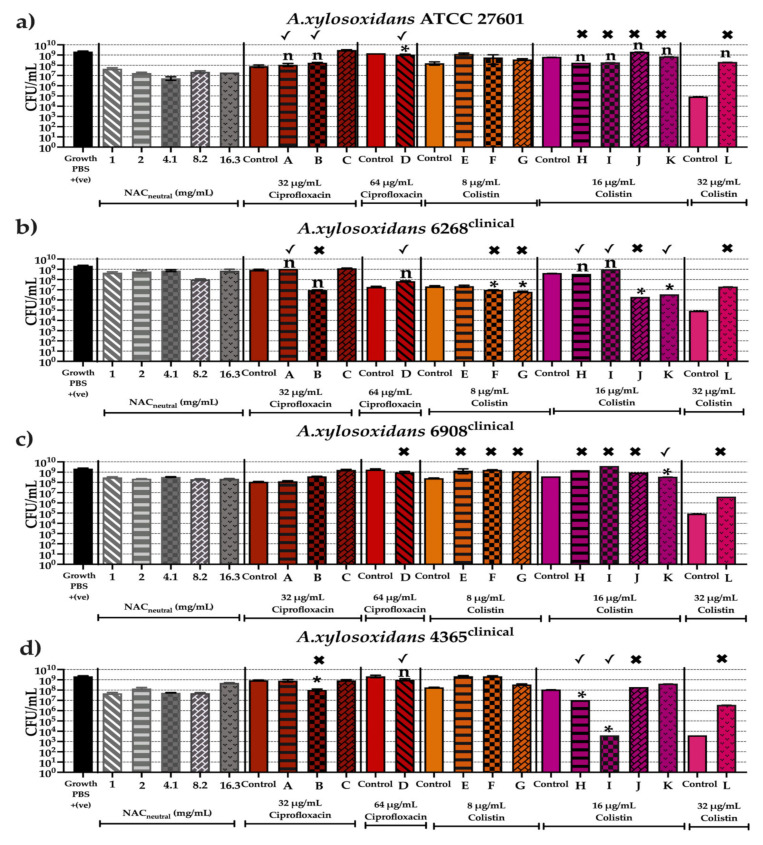
Anti-biofilm activity of screened combinations after 24 h against bacterial strains of *A. xylosoxidans* strains after 24 h using colony forming units (CFU/mL). In vitro effectiveness of combination therapies (CT) (Table 3) after 24 h on the following *A. xylosoxidans* strains (**a**) ATCC 27601, (**b**) AX 4365, (**c**) AX 6268, (**d**) AX 6908. (✓) represents combination tested does correspond to planktonic result in FICI analysis. (✖) represents combination tested does not correspond to planktonic result in FICI analysis. Statistical analyses were performed using unpaired Student’s *t*-tests (with Welch’s correction) and multiple comparison tests (Kruskal–Wallis test with Dunn’s correction). Significance cut-offs were *p* > 0.05 (*n*), *p ≤* 0.05 (*). Data represent an average of *n* = 3 biological replicates.

**Figure 8 antibiotics-10-01176-f008:**
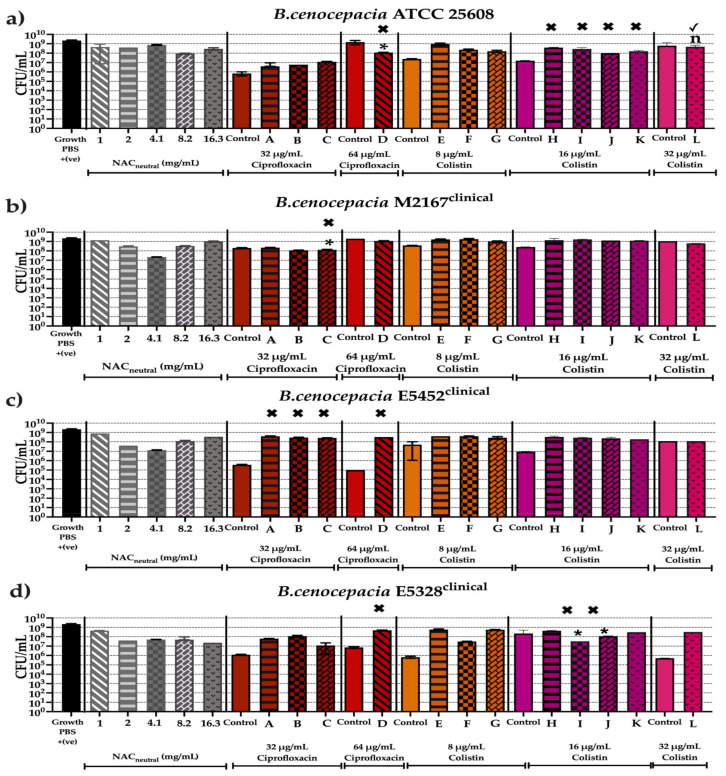
Anti-biofilm activity of screened combinations after 24 h against bacterial strains of *B. cenocepacia* using colony forming units (CFU/mL). In vitro effectiveness of combination therapies (CT) (Table 3) after 24 h on following *B. cenocepacia* strains (**a**) ATCC 25608, (**b**) M2167, (**c**) E5452, (**d**) E5328. (✓) represents combination tested does correspond to planktonic result in FICI analysis. (✖) represents combination tested does not correspond to planktonic result in FICI analysis. Statistical analyses were performed using unpaired Student’s *t*-tests (with Welch’s correction) and multiple comparison tests (Kruskal–Wallis test with Dunn’s correction). Significance cut-offs are as follows *p* > 0.05 (*n*), *p ≤* 0.05 (*). Data represent an average of *n* = 3 biological replicates.

**Figure 9 antibiotics-10-01176-f009:**
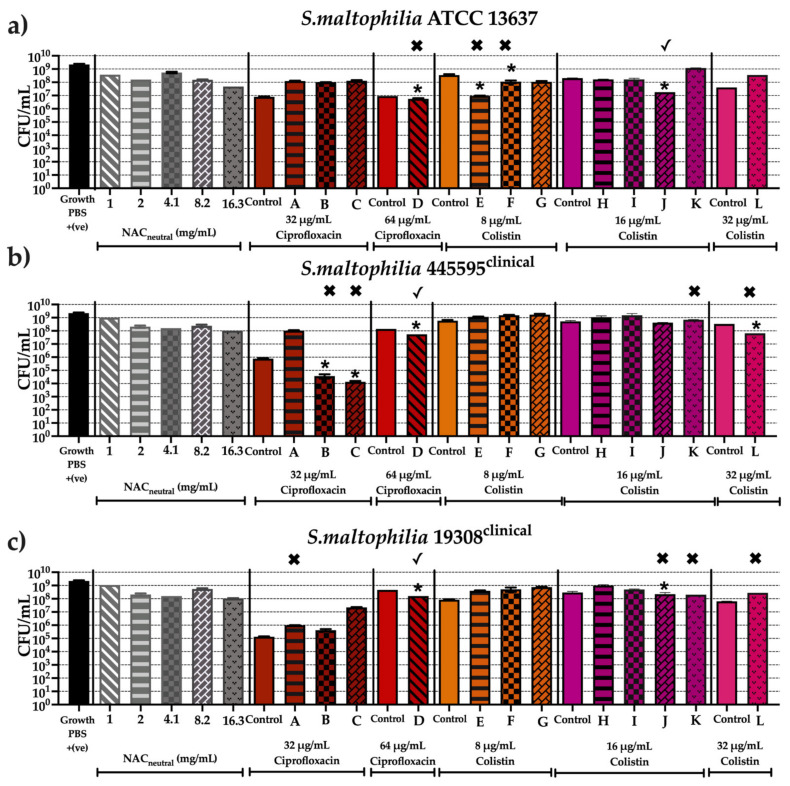
Anti-biofilm activity of screened combinations after 24 h against bacterial strains of *S. maltophilia* using colony forming units (CFU/mL). In vitro effectiveness of combination therapies (CT) (Table 3) after 24 h on following *S. maltophilia* strains (**a**) ATCC 13637, (**b**) 445595, (**c**) 19308. (✓) represents combination tested does correspond to planktonic result in FICI analysis. (✖) represents combination tested does not correspond to planktonic result in FICI analysis. Statistical analyses were performed using unpaired Student’s *t*-tests (with Welch’s correction) and multiple comparison tests (Kruskal–Wallis test with Dunn’s correction). Significance cut-offs are as *p* > 0.05 (*n*), *p ≤* 0.05 (*). Data represent an average of *n* = 3 biological replicates.

**Table 1 antibiotics-10-01176-t001:** Source and antibiotic profiles of bacterial species used.

Bacteria	Strain	Source	Antibiotic Profile				
			TZP110	CAZ30	MEM10	CIP5	SXT25
*B. cenocepacia*	ATCC 25608^TM^	ATCC (Manassas, Virginia) (Incision wound)	NA	S	S	NA	S
	M2167 ^+^	CF Sputum ^(+)^	NA	R	R	NA	R
	E5452 ^+^	CF Sputum ^(+)^	NA	S	R	NA	R
	E5328 ^+^	CF Sputum ^(+)^	NA	S	S	NA	R
*S. maltophilia*	ATCC 13637^TM^	ATCC (Oropharyngeal region)	NA	NA	NA	NA	S
	445595 ^	CF Sputum ^	NA	NA	NA	NA	S
	19308 ^	CF Sputum ^	NA	NA	NA	NA	R
*A. xylosoxidans* ^#^	ATCC 27601^TM^	ATCC (Ear discharge)	S	I	S	R	R
	6268 ^	CF Sputum ^	S	I	S	R	R
	6908 ^	CF Sputum ^	S	I	S	R	R
	4365 ^	CF Sputum ^	NA	NA	NA	NA	R

^+^ Microbiology department, Westmead Hospital, Sydney, Australia, ^ Respiratory clinic, Royal Prince Alfred Hospital, Sydney, Australia. ^#^ Antibiotic susceptibility testing performed using VITEK2 per manufacturer’s instructions, using CLSI breakpoints for “other non-enterobacterales” (1). No breakpoints exist for colistin. Antibiotics tested: TZP100, Piperacillin-Tazobactam 110 μg/mL; CAZ30, Ceftazidime, 30 µg/mL; MEM10, Meropenem 10 μg/mL; CIP5, Ciprofloxacin 5 μg/mL; SXT25, Trimethoprim-Sulfamethoxazole 1.25/23.75 μg/mL. Breakpoints are represented as S: sensitive, I: intermediate R: resistance, NA: not applicable.

**Table 2 antibiotics-10-01176-t002:** Minimum Inhibitory concentrations (MIC) of bacterial strains used in study. Testing performed using broth microdilution method in planktonic bacteria.

	Bacteria	Neutral NAC (mg/mL)	Ciprofloxacin (μg/mL)	Colistin (μg/mL)	Tobramycin (μg/mL)
*A. xylosoxidans*	ATCC 27601^TM^	>16.3	32	>128	>128
	6268	>16.3	32	32	>128
	6908	>16.3	32	8	>128
	4365	>16.3	32	>128	>128
*B. cenocepacia*	ATCC 25608 ^TM^	>16.3	64	16	>128
	M2167	>16.3	64	>128	>128
	E5452	>16.3	64	64	>128
	E5328	>16.3	64	>128	>128
*S. maltophilia*	ATCC 13637 ^TM^	>16.3	16	32	64
	445595	>16.3	64	64	>128
	19308	>16.3	32	64	64

**Table 3 antibiotics-10-01176-t003:** Combinations tested with letter references used in text. Combinations selected based on synergistic and additive effects observed during FICI analysis.

Letter Reference	Neutral NAC (mg/mL)	Antibiotic (μg/mL)
A	2	32 μg/mL Ciprofloxacin
B	4.1	
C	8.2	
D	1	64 μg/mL Ciprofloxacin
E	2	8 μg/mL Colistin
F	4.1	
G	8.2	
H	2	16 μg/mL Colistin
I	4.1	
J	8.2	
K	16.3	
L	16.3	32 μg/mL Colistin

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
