# Peer review of "Effect of N-Acetylcysteine in Combination with Antibiotics on the Biofilms of Three Cystic Fibrosis Pathogens of Emerging Importance"

_antibiotics, 2021, doi:10.3390/antibiotics10101176_

Round 1

Reviewer 1 Report

In this work A. Aiyer et. al analyze synergy between antibiotics (ciprofloxacin, colistin and tobramycin) when combined with N-acetylcysteine in its neutral pH form. Overall work is really well written, performed analysis well described and obtained results interesting. Some specific comments below.

Lines 1-4: Please remove inverted commas

Lines 7-13: Please standardize, e.g. city, University..

Line 88: “.” Before We aim.

e.g. Line 98: ATCC 25608 (in Table 1 it is ATCC 25608TM, while in Table 2 ATCC 25608). Please standardize.

Line 100: S. maltophilia in one line.

Figure 1, Figure 2: In my opinion bar graphs are too small, while descriptions take too much space on Figures. Therefore, Figures are illegible.

Lines 190-196, 232-238, 239-245 etc. : please format the description according to the rest of manuscript.

Line 546: MIC abbreviation description already in line 81. Please standardize abbreviations through the manuscript.

References: bacteria in italics.

Reviewer 2 Report

The manuscript titled “Effect of N-acetylcysteine in combination with antibiotics on the biofilms of three cystic fibrosis pathogens of emerging importance” reports on a study aimed at identifying potential synergistic combinations between three different antimicrobial agents and N-acetylcysteine against three microorganisms known to infect patients with cystic fibrosis. Interestingly, synergistic assays were conducted in planktonic and biofilm modes of growth and using different strains of the three microorganisms.

The identification of non-antibiotic molecules able to improve antibacterial activity of known antibiotics represents a promising strategy to counteract the development of AMR.

N-acetylcysteine is a known anti-biofilm agent able to synergize with different antibacterial agents, taking away novelty from this research. However, the manuscript is well-written. Although the obtained data are not exciting, they highlight the importance to investigate antibacterial activity also in microorganisms growing in biofilm conditions, strengthening this concept for the future.

Reviewer 3 Report

Major issues

My major concern is that the main conclusion of this study (i.e. that planktonic FICI analysis does not necessarily translate to reduction of bacterial loads in biofilms) is not new and could have been foreseen by the authors. It is well-known that MIC values of antibiotics often differ between planktonic and biofilm cultures, where cells grown in a biofilm often exhibit higher resistance to antibiotics and thus a higher MIC. Nevertheless, the authors chose to perform all experiments to determine NAC-antibiotic combinations with potential synergistic effects (shown in Figures 1-3) only on planktonic cultures and not on biofilm cultures. 
As it is well-known that the majority of CF lung infections occur in (polymicrobial) biofilms, it would be much more clinically relevant to look for combinations of NAC and an antibiotic that have a synergistic effect against biofilm cultures rather than planktonic cultures. To do this, the authors should have assessed the effect on growth of NAC alone, antibiotics alone and NAC + antibiotics in biofilm cultures, instead of performing the experiments on planktonic cultures and hoping that results would translate to biofilm cultures.
A second major concern is that the figures are in general difficult to interpret and not at all visually appealing. The lack of color use in most figures is strange, given that Antibiotics is an online journal where no fees apply for full-color figures. To be considered for publication, the design and execution of the figures needs to be improved. For example, in Figure 2, the bars are very small and the differences between the different ‘fills’ representing different antibiotic concentrations are very hard to distinguish. In my opinion, this requires too much effort from the reader to deduce the antibiotic concentrations resulting in significant growth reduction. Another example is Figure 3, where x and y-axis labels are included for every sub-graph, causing them to be in such a small font that it is difficult to read.

Minor issues

The introduction does not mention that Bcc bacteria are well-known for their intrinsic resistance to polymyxin antibiotics (including colistin), which I consider to be relevant given that colistin is one of the three antibiotics tested in this study. This could also be mentioned in the discussion.
Lines 74-75: the sentence suggests that synergy of NAC and colistin has been observed in Bcc, which is not the case.
Table 1: please list the source for the ATCC strains. Are these CF isolates or not?
Lines 268-288: I suggest to remove the individual OD values of each concentration to improve readability of this section. If exact OD values are important, they should be easy to discern from the supplementary material.

Round 2

Reviewer 3 Report

The authors have responded adequately to each of my comments. They have significantly improved the quality of the figures by adding color and splitting up several multipanel figures which greatly improves legibility. They have added the info requested regarding the strain sources and have included an additional statement in the discussion to better clarify the rationale for the study design.